# On the Importance of Architectures and Hyperparameters for Fairness in Face Recognition

**Rhea Sukthanker**[1][*], **Samuel Dooley**[2][*], **John P. Dickerson**[2,3], **Colin White**[4], **Frank Hutter**[1,5]
**Micah Goldblum**[6]

[1]University of Freiburg, [2]University of Maryland, [3]ArthurAI, [4]Abacus.AI,
[5]Bosch Center for AI, [6]New York University

## Abstract

Face recognition systems are used widely but are known to exhibit bias across a range of sociodemographic dimensions, such as gender and race. An array of works proposing pre-processing, training, and post-processing methods have failed to close these gaps. Here, we take a very different approach to this problem, identifying that both architectures and hyperparameters of neural networks are instrumental in reducing bias. We first run a large-scale analysis of the impact of architectures and training hyperparameters on several common fairness metrics and show that the implicit convention of choosing high-accuracy architectures may be suboptimal for fairness. Motivated by our findings, we run the first neural architecture search for fairness, jointly with a search for hyperparameters. We output a suite of models which Pareto-dominate all other competitive architectures in terms of accuracy and fairness. Furthermore, we show that these models transfer well to other face recognition datasets with similar and distinct protected attributes. We release our code and raw result files so that researchers and practitioners can replace our fairness metrics with a bias measure of their choice.

## 1 Introduction

Face recognition is regularly deployed across the world by government agencies for tasks including surveillance, employment, and housing decisions. However, recent studies have shown that face recognition systems exhibit disparity in accuracy based on race and gender [1, 2, 3, 4]. While existing methods for de-biasing face recognition systems use a fixed neural network architecture and training hyperparameters, we instead ask a fundamental question which has received little attention: *does model-bias stem from the architecture and hyperparameters?* We further exploit the extensive research in the fields of neural architecture search (NAS) [5] and hyperparameter optimization (HPO) [6] to search for models that achieve a desired trade-off between bias and accuracy.

In this work, we take the first step towards answering these questions. To this end, we conduct the first large-scale analysis of the relationship between hyperparameters, architectures, and bias. We train a diverse set of 29 architectures, ranging from ResNets [7] to vision transformers [8, 9] to Gluon Inception V3 [10] to MobileNetV3 [11] on CelebA [12], for a total of 88 493 GPU hours. We train each of these architectures across different head, optimizer, and learning rate combinations. Our results show that different architectures learn different inductive biases from the same dataset. We conclude that the implicit convention of choosing the highest-accuracy architectures can be detrimental to fairness, and suggest that architecture and hyperparameters play a significant role in determining the fairness-accuracy tradeoff.

Next, we exploit this observation in order to design architectures with a better fairness-accuracy tradeoff. We initiate the study of NAS for fairness; specifically, we run NAS+HPO to jointly

---

[*]Equal contribution. Email to: sukthank@cs.uni-freiburg.de, sdooley1@cs.umd.edu.

2022 Trustworthy and Socially Responsible Machine Learning (TSRML 2022) co-located with NeurIPS 2022.

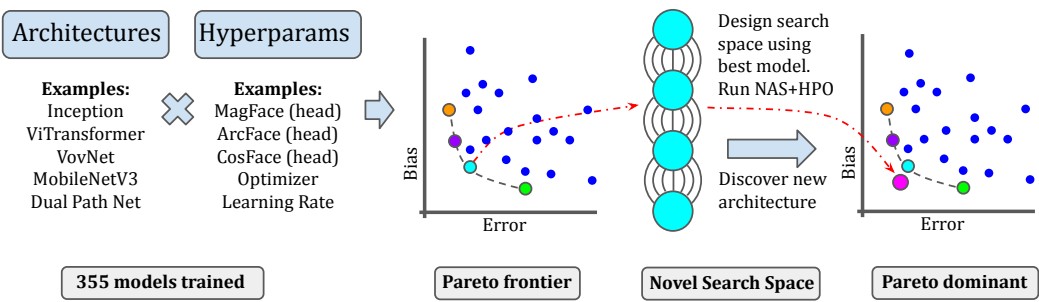

Figure 1: Overview of our methodology.

optimize fairness and accuracy. To tackle this problem, we construct a search space based on the highest-performing architecture from our analysis. We use the Sequential Model-based Algorithm Configuration method (SMAC [13]), for multi-objective architecture and hyperparameter search. We discover a Pareto frontier of face recognition models that outperform existing state-of-the-art models on both accuracy and multiple fairness metrics. An overview of our methodology can be found in Figure 1. We release all of our code and raw results at https://github.com/dooleys/FR-NAS so that users can adapt our work to any bias measure of their choice.

**Our contributions.** We summarize our main contributions below:

- We run a large-scale study of 29 architectures from ViT to Xception, each trained across a variety of hyperparameters, totalling 88 493 GPU hours. Our analysis shows that there is a distinct trade-off between accuracy and popular fairness metrics, such as disparity, and simply improving accuracy would not guarantee improvement on different fairness metrics.
- Motivated by the above observation, we conduct the first neural architecture search for fairness, jointly with hyperparameter optimization and optimizing for accuracy — culminating in a set of architectures which Pareto-dominate all models in a large set of modern architectures
- We show that the architectures discovered transfer across different datasets with the same (perceived gender) and different (ethnicities) protected attributes.

**Background and related work.** Face recognition tasks fall into two categories: verification and identification. *Verification* asks whether the person in a source image is the same person as in the target image; this is a one-to-one comparison. *Identification* instead asks whether a given person in a source image appears within a gallery composed of many target identities and their associated images; this is a one-to-many comparison. Novel techniques in face recognition tasks [14, 15, 16] use deep networks to extract feature representations of faces and then compare those to match individuals (with mechanisms called the *head*). We focus our analysis on identification and on examining how close images of similar identities are in the feature space of trained models.

In this work, we focus on *measuring* sociodemographic disparities across neural architectures and hyperparameter settings, and finding the Pareto frontier of face recognition performance and bias for current and novel architectures. Our work searches for architectures and hyperparameters which improve undesired disparities. A few works have applied hyperparameter optimization to mitigate bias in models for tabular data. Perrone et al. [17] recently introduced a Bayesian optimization framework to optimize accuracy while satisfying a bias constraint. The concurrent works of Schmucker et al. [18] and Cruz et al. [19] extend Hyperband [20] to the multi-objective setting and apply it to fairness. To the best of our knowledge, no prior work uses any AutoML technique (NAS, HPO, or joint NAS and HPO) to design fair face recognition models, and no prior work uses NAS to design fair models for any application. For additional related work, see Appendix A.

## 2 A Large-Scale Analysis of Architectures and Fairness

**Experimental Setup.** We train and evaluate each model configuration on a gender-balanced subset of the CelebA dataset [12]. While this work analyzes phenotypal metadata (perceived gender), the reader should not interpret our findings as a social lens of what these demographic groups mean inside society. We guide the reader to Hamidi et al. [21] and Keyes [22]. We use the following training

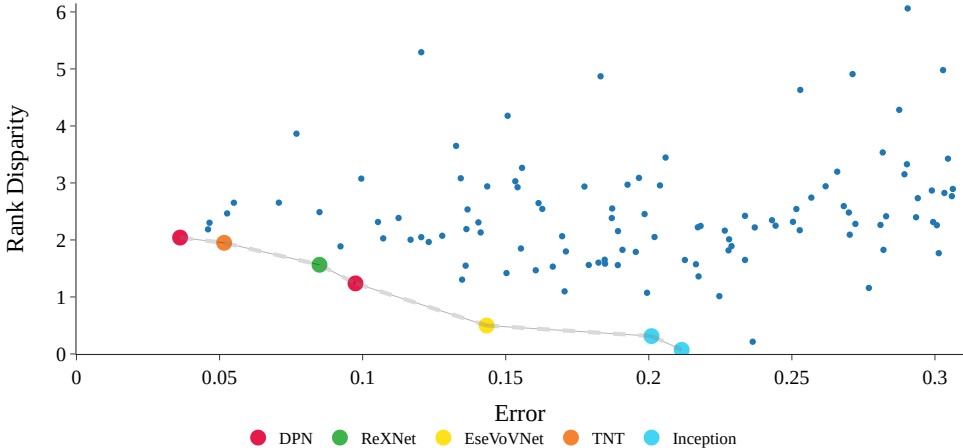

Figure 2: Error-Rank Disparity Pareto front of the architectures with lowest error (< 0.3). Models in the lower left corner are better. The Pareto front is notated with a dashed line. Other points are architecture and hyperparameter combinations which are not Pareto-optimal. DPN, ReXNet, EseVovNet, TNT, and Inception architectures are Pareto-optimal.

pipeline – ultimately conducting 355 training runs with different combinations of 29 architectures from the Pytorch Image Model (`timm`) database [23] and hyperparameters. For each model, we use the default learning rate and optimizer that was published with that model. We then conduct a training run with these hyperparameters and each of three heads, ArcFace [14], CosFace [15], and MagFace [16]. Next, we use that default learning rate with both AdamW [24] and SGD optimizers (again with each head). Finally, we also conduct training routines with AdamW and SGD with unifed learning rates. In total, we run a single architecture between 9 and 13 times. All other hyperparameters were the same for each model training run.

**Evaluation procedure.**

We evaluate performance via *Error* and use a common fairness metric in face recognition, *rank disparity*, which is explored in the NIST FRVT [25]. To compute the rank of a given sample, we ask how many images of a different identity are closer to it in feature space. We define this index as the *Rank* of a given image. Thus, *Rank(image) = 0* if and only if *Error(image) = 0*; *Rank(image) > 0* if and only if *Error(image) = 1*. We examine the **rank disparity** which is the absolute difference of the average ranks for each perceived gender in a dataset $\mathcal{D}$:

$$\text{Rank Disparity} = \left| \frac{1}{|\mathcal{D}_{\text{male}}|} \sum_{x \in \mathcal{D}_{\text{male}}} \text{Rank}\,(x) - \frac{1}{|\mathcal{D}_{\text{female}}|} \sum_{x \in \mathcal{D}_{\text{female}}} \text{Rank}(x) \right|.$$

**Results and Discussion.**   By plotting the performance of each training run with the error on the $x$-axis and rank disparity on the $y$-axis in Figure 2, we can easily conclude two main points. First, optimizing for error does not also optimize for fairness, and second, different architectures have different fairness properties.

On the first point, a search for architectures and hyperparameters which have high performance on traditional metrics does not translate to high performance on fairness metrics. We see that within models with lowest error – those models which are most interesting to the community – there is low correlation between error and rank disparity ($\rho = -.113$ for models with error < 0.3). In Figure 2, we see that Pareto optimal models are versions of DPN, TNT, ReXNet, VovNet, and ResNets (in increasing error and decreasing fairness). We conclude that both architectures and hyperparameters play a significant role in determining the accuracy and fairness trade-off, motivating their joint optimization in Section 3.

Additionally, we observe that the Pareto curve is dependent upon what fairness metric we consider. For example, in Figure 3, we demonstrate that a very different set of architectures are Pareto optimal

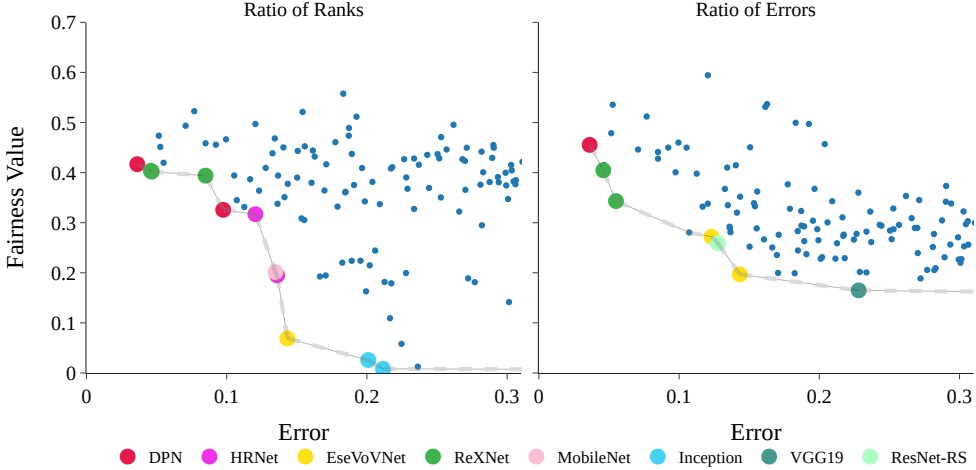

Figure 3: Depending on the fairness metric, different architectures are Pareto-optimal. On the left, we plot the metric Ratio of Ranks which admit DPN, ReXNet, HRNet, MobileNet, EseVovNet, and Inceptions as Pareto-optimal. On the right, we plot the metric Ratio of Errors where DPN, ReXNet, EseVovNet, ResNet-RS, and VGG19 are Pareto-optimal.

if instead of rank disparity (rank difference between perceived genders) we consider the ratio of ranks between the two perceived genders or the ratio of the errors. Specifically, on the ratio of ranks metric, the Pareto frontier contains versions of HRNet, MobileNet, VovNet, and ResNet whereas the Pareto frontier under the ratio of errors metric includes versions of NesT, ResNet-RS, and VGG19.

Further, different architectures exhibit different optimal hyperparameters. For example, the Xception65 architecture finds SGD with ArcFace and AdamW with ArcFace are Pareto-optimal whereas the Inception-ResNet architecture finds MagFace and CosFace optimal with SGD. This illustrates the care that needs to be taken when choosing a model – optimizing architectures and hyperparameters for error alone will not lead to fair models.

Finally, existing architectures and hyperparameters do not yield models which simultaneously exhibit both low error and low disparity. For example, in Figure 2 there is a significant area under the Pareto curve. While there are models with very low error, in order to improve the disparity metric, one must sacrifice significant performance. However, in Section 3, we will see that our joint NAS+HPO experiments for rank disparity ultimately find a model convincingly in the area to the left of this Pareto curve – that is, we find a model with low error *and* disparity.

## 3 Joint NAS+HPO for Fairness

In this section, we employ joint NAS+HPO to find better architectures. Inspired by our findings on the importance of architectures and hyperparameters for fairness in Section 2, we initiate the first joint NAS+HPO study for fairness in face recognition. We start by describing our search space and search strategy. We then present a comparison between the architectures obtained from multi-objective joint NAS+HPO and the handcrafted image classification models studied in Section 2. We conclude that our joint NAS+HPO indeed discovers simultaneously accurate and fair architectures.

### 3.1 Search Space Design

We design our search space based on our analysis in Section 2 namely Dual Path Networks [26] due to its strong trade-off between rank disparity and accuracy as seen in Figure 2. We choose three categories of hyperparameters for NAS+HPO: architecture head/loss, optimizer, and learning rate.

**Architecture Search Space Design.** Dual Path Networks [26] for image classification share common features (ResNets [27]) while possessing the flexibility to explore new features [28] through a dual path architecture. We replace the repeating `1x1_conv-3x3_conv-1x1_conv` block with a

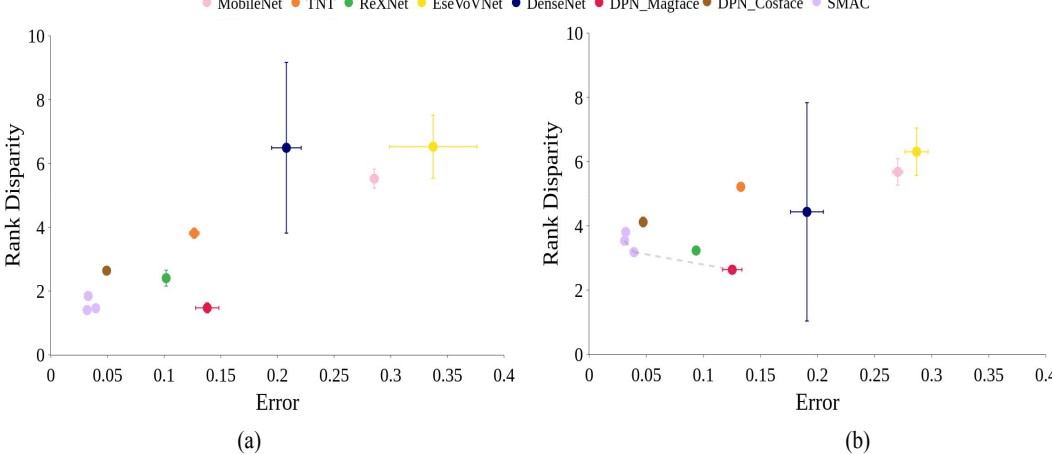

Figure 4: Pareto front of models discovered by SMAC and rank-1 models from `timm` for *(a)* validation and *(b)* test sets on CelebA averaged over 4 seeds. SMAC models Pareto-dominate top performing `timm` models ($Error < 0.1$).

simple recurring searchable block. Furthermore, we stack multiple such searched blocks to closely follow the architecture of Dual Path Networks. We have nine possible choices for each of the three operations in the DPN block. The choices include a vanilla convolution, a convolution with pre-normalization and a convolution with post-normalization. To summarize, our search space consists of a choice among 81 different architecture types, 3 different head types, 3 different optimizers (discrete hyperparameters) and a possibly infinite number of choices for the continous learning rate.

## 3.2 Search Strategy

We navigate the search space defined in Section 3.1 using Black-Box-Optimization (BBO). We want our BBO algorithm to support the following important techniques:

**Multi-fidelity optimization.** A single function evaluation for our use-case corresponds to training a deep neural network on a given dataset. This is generally quite expensive for traditional deep neural networks on moderately large datasets. Hence we would like to use cheaper approximations to speed up the black-box algorithm with multi-fidelity optimization techniques [29, 20, 30], e.g., by evaluating many configurations based on short runs with few epochs and only investing more resources into the better-performing ones.

**Multi-objective optimization.** We want to observe a trade-off between the accuracy of the face recognition system and the fairness objective of choice (rank disparity). Hence, our joint NAS+HPO algorithm needs to support multi-objective optimization [31, 32, 33]. The SMAC3 package [13] offers a robust and flexible framework for Bayesian Optimization with few evaluations. SMAC3 offers a SMAC4MF facade for *multi-fidelity optimization* to use cheaper approximations for expensive deep learning tasks like ours. We choose ASHA [29] for cheaper approximations with the initial and maximum fidelities set to 25 and 100 epochs, respectively, and $\eta = 2$. Every architecture-hyperparameter configuration evaluation is trained using the same training pipeline as in Section 2. For the sake of simplicity, we use ParEGO [32] for *multi-objective optimization* with $\rho$ set to 0.05.

## 3.3 Results

We follow the evaluation scheme of Section 2 to compare models discovered by joint NAS+HPO with the handcrafted models. In Figure 4, we compare the set of models discovered by joint NAS+HPO vs. the models on the Pareto front from Section 2. We train each model for 4 seeds to study the robustness of error and disparity. As seen in Figure 4, we Pareto-dominate all other models with above random accuracy on the validation set. On the test set, we still Pareto-dominate all competitive models (with $Error < 0.1$), but due to differences between the two dataset splits, one of the original configurations (DPN with Magface) also becomes Pareto-optimal. However, the error of this architecture is 0.13, which is significantly higher than the the best original model (0.05) and the SMAC models (0.03-0.04). Furthermore, from Figure 4 it is also apparent that some models such as VoVNet and DenseNet show

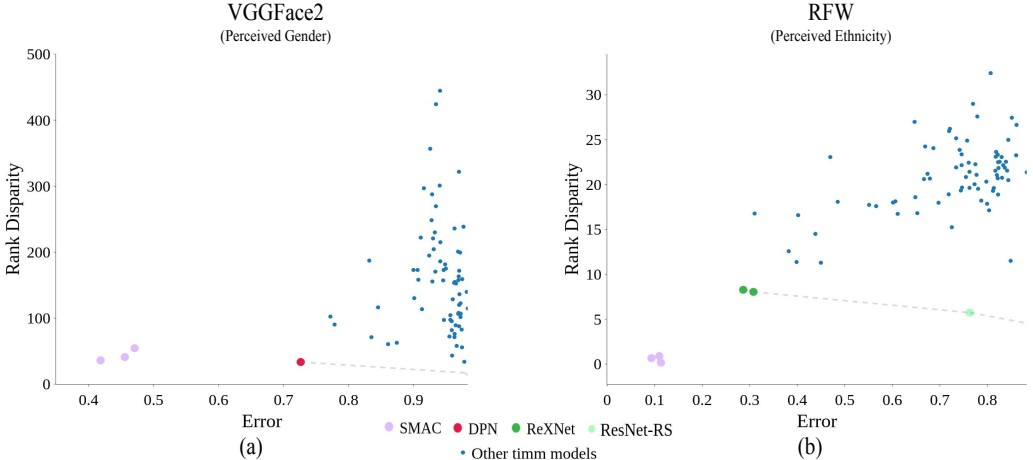

Figure 5: Pareto front of the models on *(a)* the VGGFace2 test set with perceived gender as the protected attribute and *(b)* the RFW test set with perceived ethnicity as the protected attribute. The SMAC models discovered by joint NAS+HPO Pareto-dominate the `timm` models.

very large standard errors across seeds. Hence, it becomes very important to also study the robustness of the models across seeds along with the accuracy and disparity Pareto front.

### 3.3.1 Transfer across Face Recognition Datasets

Inspired by our findings on the CelebA dataset, we now study the accuracy-disparity trade-off of the models studied in Section 2 and the searched models from Section 3 on two different datasets. The first face recognition dataset we use is VGGFace2 [34], which is based on the same protected attribute (perceived gender) that has served as the focus of our study. The second dataset, Racial Faces in the Wild (RFW) [35], consists of four different racial identities: Caucasian, Indian, Asian, and African. We compute the rank disparity within different *ethnicities*, i.e., a different attribute than the *perceived gender* studied in previous sections. With this dataset, we aim to study the generalization of the fair representations learned by the models across a different protected attribute. However, we caution the reader that the labels of these datasets rely on socially constructed concepts of gender presentation and ethnicity. The intention here is to study how the models discovered by SMAC generalize to these datasets and compare to the other handcrafted `timm` [23] architectures.

To evaluate our models on these datasets, we directly transfer our models to the two test sets. That is, we use the models trained on CelebA, without re-training or fine-tuning the models on the new datasets. As observed in Figure 5, the models discovered using joint NAS+HPO still remain Pareto-optimal on both datasets. In the case of VGGFace2, the models found by SMAC are the only ones to have an error below 0.5, where the next-best model has an error above 0.7. In the case of RFW, the models found by SMAC have considerably lower rank disparity *and* error than the standard models studied in Section 2. This might be due to the optimized architectures learning representations that are intrinsically fairer than those of standard architectures, but it requires further study to test this hypothesis and determine in precisely which characteristics these architectures differ.

## 4 Conclusion and Future Work

We conducted the first large-scale analysis of the relationship among hyperparameters and architectural properties, and accuracy, bias, and disparity in predictions. We expect the future work in this direction to focus on studying different multi-objective algorithms [36, 37] and NAS techniques [38, 39, 40] to search for inherently fairer models. Further, it would be interesting to study how the properties of the architectures discovered translate across different demographics and populations. Another potential direction of future work is including priors and beliefs about fairness in the society from experts to further improve and aid NAS+HPO methods for fairness by integrating expert knowledge. Finally, given the societal importance of fairness, it would be interesting to study how our findings translate to real-life face recognition systems under deployment.

## Acknowledgments

This research was partially supported by the following sources: NSF CAREER Award IIS-1846237, NSF D-ISN Award #2039862, NSF Award CCF-1852352, NIH R01 Award NLM-013039-01, NIST MSE Award #20126334, DARPA GARD #HR00112020007, DoD WHS Award #HQ003420F0035, ARPA-E Award #4334192; TAILOR, a project funded by EU Horizon 2020 research and innovation programme under GA No 952215; the German Federal Ministry of Education and Research (BMBF, grant RenormalizedFlows 01IS19077C); the Deutsche Forschungsgemeinschaft (DFG, German Research Foundation) under grant number 417962828; the European Research Council (ERC) Consolidator Grant "Deep Learning 2.0" (grant no. 101045765). Funded by the European Union. Views and opinions expressed are however those of the author(s) only and do not necessarily reflect those of the European Union or the ERC. Neither the European Union nor the ERC can be held responsible for them.

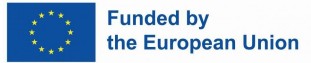

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

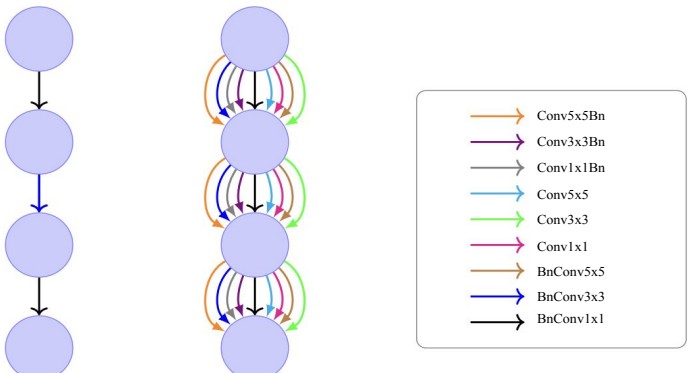

Figure 6: DPN block (left) vs. our searchable block (right).

# A   Additional Related Work

**Face Recognition.**   Face recognition tasks fall into two categories: verification and identification. *Verification* asks whether the person in a source image is the same person as in the target image; this is a one-to-one comparison. *Identification* instead asks whether a given person in a source image appears within a gallery composed of many target identities and their associated images; this is a one-to-many comparison. Novel techniques in face recognition tasks, such as ArcFace [14], CosFace [15], and MagFace [16], use deep networks to extract feature representations of faces and then compare those to match individuals (with mechanisms called the *head*). We focus our analysis on identification and on examining how close images of similar identities are in the feature space of trained models.

**Sociodemographic Disparities in Face Recognition.**   In this work, we focus on *measuring* the sociodemographic disparities across neural architectures and hyperparameter settings, and finding the Pareto frontier of face recognition performance and bias for current and novel architectures. Our work searches for architectures and hyperparameters which improve the undesired disparities. Previous work focuses on "fixing" unfair systems and can be split into three (or arguably four [41]) focus areas: preprocessing [e.g., 42, 43, 44, 45], inprocessing [e.g., 46, 47, 48, 49, 50, 45, 51, 52, 53, 54], and post-processing [e.g., 55, 56].

**Neural Architecture Search (NAS) and Hyperparameter Optimization (HPO).** A few works have applied hyperparameter optimization to mitigate bias in models for tabular datasets. Perrone et al. [17] recently introduced a Bayesian optimization framework to optimize accuracy of models while satisfying a bias constraint. The concurrent works of Schmucker et al. [18] and Cruz et al. [19] extend Hyperband [20] to the multi-objective setting and show its applications to fairness. The former was later extended to the asynchronous setting [29]. Lin et al. [57] proposes de-biasing face recognition models through model pruning. However, they consider just two architectures and just one set of hyperparameters. To the best of our knowledge, no prior work uses any AutoML technique (NAS, HPO, or joint NAS and HPO) to design fair face recognition models, and no prior work uses NAS to design fair models for any application.

## A.1   Search Space Design

We design our search space based on our analysis in Section 2. In particular, our search space is inspired by Dual Path Networks [26] due to its strong trade-off between rank disparity and accuracy as seen in Figure 2.

**Hyperparameter Search Space Design.**   We choose three categories of hyperparameters for NAS+HPO: the architecture head/loss, the optimizer, and the learning rate, depicted in Table 1.

**Architecture Search Space Design.**   Dual Path Networks [26] for image classification share common features (ResNets [27]) while possessing the flexibility to explore new features [28] through a dual path architecture. We replace the repeating `1x1_conv-3x3_conv-1x1_conv` block with a simple recurring searchable block depicted in Figure 6. Furthermore, we stack multiple such

Table 1: Searchable hyperparameter choices.

| Hyperparameter | Choices |
|---|---|
| Architecture Head/Loss | MagFace, ArcFace, CosFace |
| Optimizer Type | Adam, AdamW, SGD |
| Learning rate (conditional) | Adam/AdamW $\rightarrow [1e-4, 1e-2]$, SGD $\rightarrow [0.09, 0.8]$ |

Table 2: Operation choices and definitions.

| Operation | Definition |
|---|---|
| BnConv1x1 | Batch Normalization $\rightarrow$ Convolution with 1x1 kernel |
| Conv1x1Bn | Convolution with 1x1 kernel $\rightarrow$ Batch Normalization |
| Conv1x1 | Convolution with 1x1 kernel |
| BnConv3x3 | Batch Normalization $\rightarrow$ Convolution with 3x3 kernel |
| Conv3x3Bn | Convolution with 3x3 kernel $\rightarrow$ Batch Normalization |
| Conv3x3 | Convolution with 3x3 kernel |
| BnConv5x5 | Batch Normalization $\rightarrow$ Convolution with 5x5 kernel |
| Conv5x5Bn | Convolution with 5x5 kernel $\rightarrow$ Batch Normalization |
| Conv5x5 | Convolution with 5x5 kernel |

searched blocks to closely follow the architecture of Dual Path Networks. We have nine possible choices for each of the three operations in the DPN block as depicted in Table 2. The choices include a vanilla convolution, a convolution with pre-normalization and a convolution with post-normalization.

## B Further Details on Experimental Design and Results

### B.1 Experimental Setup

The list of the models we study from `timm` are: `coat_lite_small` [58], `convit_base` [59], `cspdarknet53` [60], `dla102x2` [61], `dpn107` [26], `ese_vovnet39b` [62], `fbnetv3_g` [63], `ghostnet_100` [64], `gluon_inception_v3` [10], `gluon_xception65` [65], `hrnet_w64` [66], `ig_resnext101_32x8d` [67], `inception_resnet_v2` [68], `inception_v4` [68], `jx_nest_base` [69], `legacy_senet154` [70], `mobilenetv3_large_100` [11], `resnetrs101` [71], `rexnet_200` [72], `selecsls60b` [73], `swin_base_patch4_window7_224` [9], `tf_efficientnet_b7_ns'` [74], `'tnt_s_patch16_224` [75], `twins_svt_large` [76] , `vgg19` [77], `vgg19_bn` [77], `visformer_small` [78], `xception` and `xception65` [65].

We study at most 13 configurations per model ie 1 default configuration corresponding to the original model hyperparameters with CosFace as head. Further, we have at most 12 configs consisting of the 3 heads (CosFace, ArcFace, MagFace) $\times$ 2 learning rates(0.1,0.001) $\times$ 2 optimizers (SGD, AdamW). All the other hyperparameters are held constant for training all the models. All model configurations are trained with a total batch size of 64 on 8 RTX2080 GPUS for 100 epochs each.

### B.2 Obtained architectures and hyperparameter configurations from Black-Box-Optimization

In Figure 7 we present the architectures and hyperparameters discovered by SMAC. Particularly we observe that `conv 3x3` followed `batch norm` is a preferred operation and CosFace is the preferred head/loss choice.

### B.3 Analysis of the Pareto-Front of different Fairness Metrics

In this section, we include additional plots that support and expand on the main paper. Primarily, we provide further context of the Figures in the main body in two ways. First, we provide replication

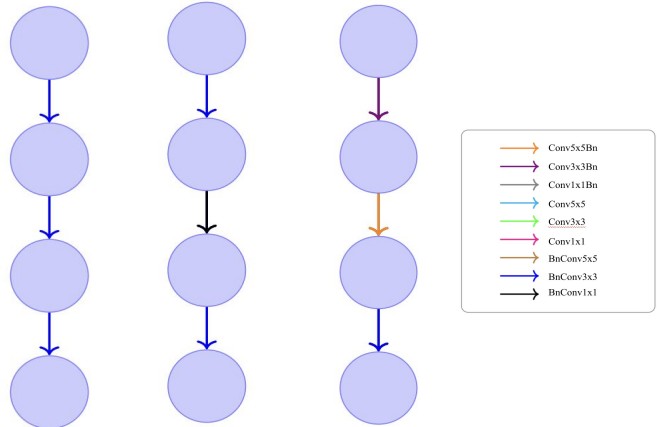

Figure 7: SMAC discovers the above building blocks with (a) corresponding to architecture with CosFace, with SGD optimizer and learning rate of 0.2813 as hyperparamters (b) corresponding to CosFace, with SGD as optimizer and learning rate of 0.32348 and (c) corresponding to CosFace, with AdamW as optimizer and learning rate of 0.0006

Table 3: Fairness Metrics Overview

| Fairness Metric | Equation |
|---|---|
| Disparity | $\lvert Accuracy(male) - Accuracy(female)\rvert$ |
| Rank Disparity | $\lvert Rank(male) - Rank(female)\rvert$ |
| Ratio | $\lvert 1 - \frac{Accuracy(male)}{Accuracy(female)}\rvert$ |
| Rank Ratio | $\lvert 1 - \frac{Rank(male)}{Rank(female)}\rvert$ |
| Error Ratio | $\lvert 1 - \frac{Error(male)}{Error(female)}\rvert$ |

plots of the figures in the main body but for all models. Recall, the plots in the main body only show models with Error<0.3, since high performing models are the most of interest to the community. Second, we also show figures which depict other fairness metrics used in facial recognition. The formulas for these additional fairness metrics can be found in Table 3.

We replicate Figure 2 in Figure 8; Figure 3 in Figure 9; Figure 5 in Figure 10 and Figure 11. We add additional metrics with Disparity being plotted in Figure 12 and Ratio being plotted in Figure 13.

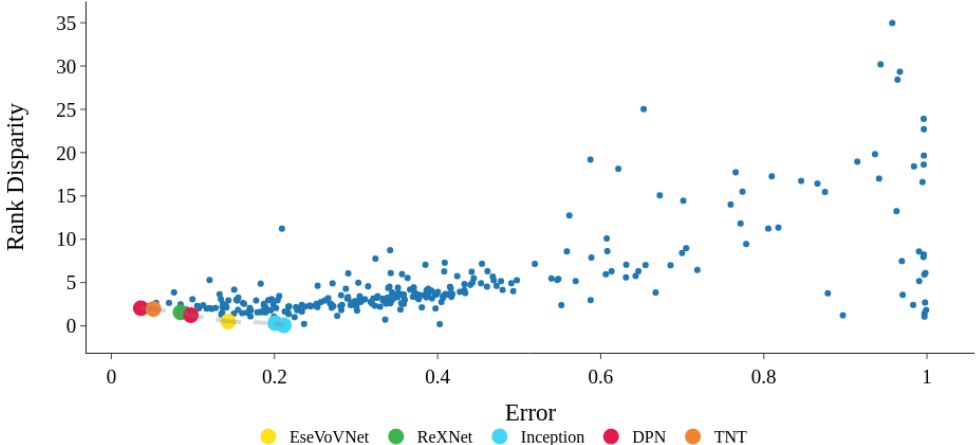

Figure 8: Replication of Figure 2 with all data points. Error-Rank Disparity Pareto front of the architectures with any non-trivial error. Models in the lower left corner are better. The Pareto front is notated with a dashed line. Other points are architecture and hyperparameter combinations which are not Pareto-dominant. DPN, ReXNet, EseVovNet, TNT, and Inception architectures are Pareto-dominant.

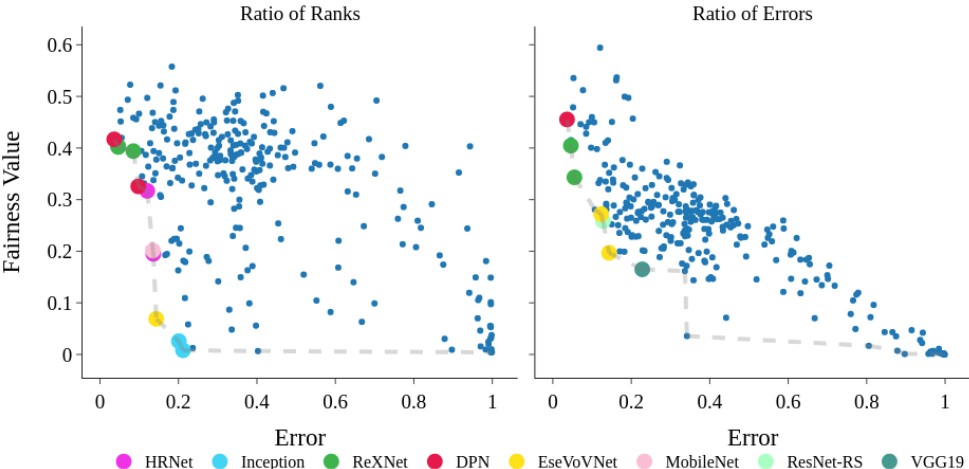

Figure 9: Replication of Figure 3 with all data points. Depending on the fairness metric, different architectures are Pareto-optimal. On the left, we plot the metric Ratio of Ranks which admit DPN, ReXNet, HRNet, MobileNet, EseVovNet, and Inceptions as Pareto-optimal. On the right, we plot the metric Ratio of Errors where DPN, ReXNet, EseVovNet, ResNet-RS, and VGG19 are architectures which are Parto-optimal

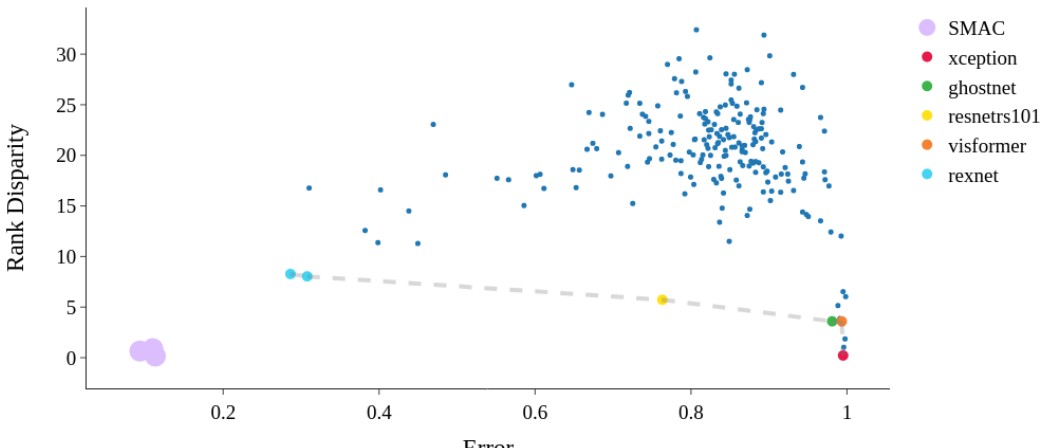

Figure 10: Replication of Figure 5 for VGGFace2 with all data points. Pareto front of the models on VGGFace2 test set with perceived gender as the protected attribute. The SMAC models discovered by joint NAS and HPO Pareto-dominate the `timm` models

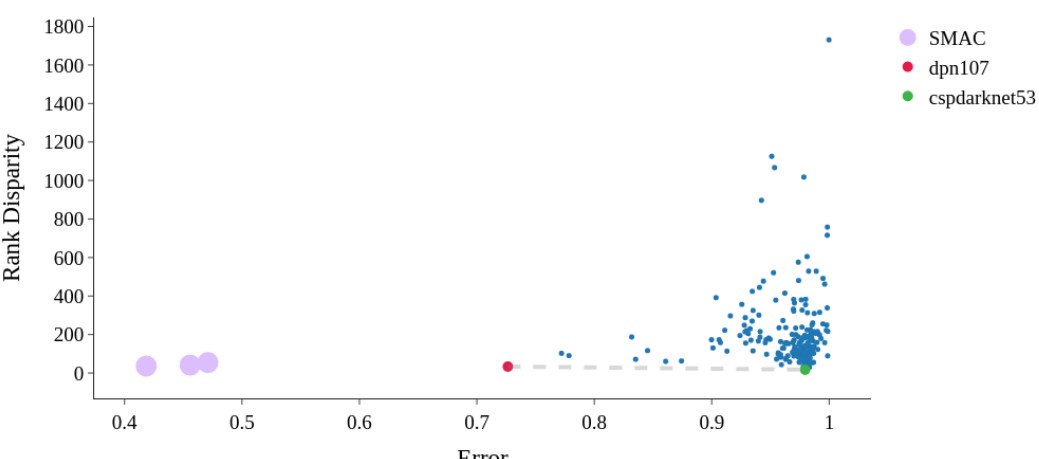

Figure 11: Replication of Figure 5 for RFW with all data points. Pareto front of the models on RFW test set with perceived ethnicity as the protected attribute. The SMAC models discovered by joint NAS and HPO Pareto-dominate the `timm` models

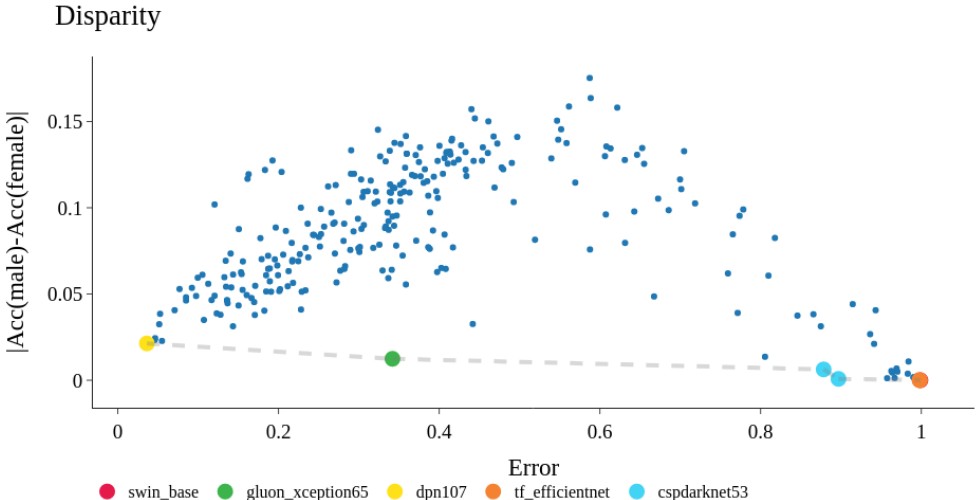

Figure 12: Extension of Figure 2 with all data points with the Disparity in accuracy metric. Error-Disparity Pareto front of the architectures with any non-trivial error. Models in the lower left corner are better. The Pareto front is notated with a dashed line. Other points are architecture and hyperparameter combinations which are not Pareto-dominant.

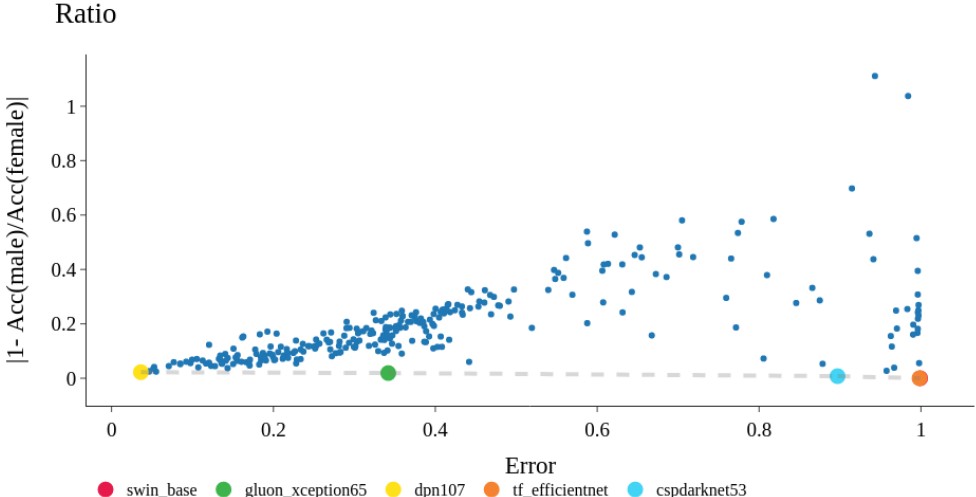

Figure 13: Extension of Figure 2 with all data points with the Ratio in accuracy metric. Error-Ratio Pareto front of the architectures with any non-trivial error. Models in the lower left corner are better. The Pareto front is notated with a dashed line. Other points are architecture and hyperparameter combinations which are not Pareto-dominant.

