# OpenReview forum: "On the Importance of Architectures and Hyperparameters for Fairness in Face Recognition"
_NeurIPS.cc/2022/Workshop/TSRML — TSRML2022_

### Official Review · Reviewer_tFw6 · 2022-10-19
**An interesting evaluation of the effect of different architectures and hyperparameters on fairness in face recognition.**

**Overall Rating:** 7

**Summary:**

The authors investigate the effect of architecture choice and hyperparameter choice on the testing error and bias for different datasets. They also apply a method to perform architecture and hyperparameter search while specifically targeting a good tradeoff between testing error and bias. The authors also discover that pareto optimal architectures and hyperparameters transfer across tasks.

The authors test on Inception, ViTransformer, VovNet, MobileNetV3 and Dual Path Net models as well as the MagFace, ArcFace, CosFace, Optimizer and Learning rate hyperparameters. They use the CelebA, VGGFace2 and Racial Faces in the Wild (RFW) datasets.

**Strengths:**

The work is relevant and timely as bias resulting from applying machine learning to critical tasks is an ongoing concern. The paper is clearly written and the results clearly presented. The experiments are extensive and of sufficient scope to derive conclusions. To the best of my knowledge the results are novel, in particular the observation that optimal hyperparameters and architectures transfer across tasks while balancing bias and testing error is very interesting.

**Weaknesses:**

The results are a bit preliminary. In particular, it would be interesting to explore the transferability of optimal hyperparameters and architectures across tasks for different datasets (I would say ~10 datasets),  for example bias in other datasets that are not of faces is also relevant. The submission doesn't propose a new algorithm and is simply an evaluation of previous algorithms, architectures etc. Furthermore, there is no intuition about why the observed tradeoff effects occur.

**Overall Recommendation:**

I recommend that this submission be accepted as it seems a sufficiently detailed empirical investigation of a relevant topic. However, I am not an expert on fairness and have set my confidence score accordingly. I cannot rule out that the same or a similar investigation has been conducted already.

**Review Confidence:**

3: The reviewer is fairly confident that the evaluation is correct

---

### Official Review · Reviewer_SLp4 · 2022-10-20
**Review for Paper35**

**Overall Rating:** 7

**Summary:**

Different from existing works which focus on proposing specially designed methods for ensuring fairness in face recognition, the paper studies the effect of architectures and hyperparameters on fairness. A set of large-scale experiments are conducted with both neural architecture search and hyperparameter search.

**Strengths:**

1. The idea of trying different model architectures and hyperparameters is novel and interesting. It potentially provides insights for which kind of backbone setting is relatively more fair than others.
2. The experiments are well-designed and results are presented appropriately.

**Weaknesses:**

1. The main takeaway message should be better summarized. The pareto optimal models shown in Figure 2 are different from Figure 3. This is a reasonable observation but what can we learn from it? It seems that the authors attempted to demonstrate that experiments with joint NAS and HPO is better. From this aspect, Section 2 is more like a motivation of the proposed method.
2. The cost (e.g., running time) for the proposed method should be compared with the methods in Figures 2,3.


**Overall Recommendation:**

The paper attempts to solve the fair training problem by jointly considering NAS and HPO. The idea is novel and may inspire future research on fair learning.

**Review Confidence:**

3: The reviewer is fairly confident that the evaluation is correct

---

### Decision · Program_Chairs · 2022-10-23

Accept